# Predictors of annual membership renewal to increase the sustainability of the Nepal National Health Insurance program: A cross-sectional survey

Gaj Bahadur Gurung *, Alessio Panza

College of Public Health Sciences, Chulalongkorn University, Bangkok, Thailand

☯ These authors contributed equally to this work.
* gaj496@gmail.com

**Data Availability Statement:** The data underlying this publication is fully available with the paper and its Supporting Information files. The data may also be accessed via an author.

## Abstract

Expanding membership coverage and retention of the National Health Insurance (NHI) programs among informal sector workers (ISWs) continues to be a significant challenge in most low and lower-middle-income countries (LMICs). The Nepal NHI program is also facing a similar problem, but to date, there are no studies that focus on identifying key predictors of annual membership renewal and retention in Nepal. This study therefore aimed to determine the predictors of intention to renew annual subscription to the NHI program among enrolled members. This cross-sectional quantitative study was part of a larger mixed-methods study conducted in three districts in Nepal. A random sample of 182 current NHI members and 61 dropped out NHI members who met the inclusion criteria were interviewed. The study's dependent variable was the intention to renew annual membership and employed univariate regression to assess the bivariate associations with the independent variables. The multivariate logistic regression examined the net effect of the independent variables on the odds of intention to renew. Our results showed that the household (HH) with high monthly income had lower odds of renewing their annual NHI membership (adjusted OR: 0.14, 95% CI: 0.03–0.58). Similarly, households (HHs) with overall health service satisfaction (adjusted OR:3.59, 95%CI: 1.23–10.43) and increased frequency of visits after NHI membership (adjusted OR: 10.09, 95% CI: 1.39–73.28) had high odds of renewing their membership. The top three dropout reasons were health services underutilization (43.3%), poor health services (26.9%), and the inadequacy of the benefits package (14.9%). Almost 64% of the respondents were willing to renew their membership upon improved services. The study found that the Nepal NHI annual membership renewal key predictors are HH income, health service quality, and health service utilization. Among these three key predictors, health service quality and service utilization were among the top three dropout reasons. The study, however, did not differentiate between moral hazards or actual service utilization, demanding further studies on the health service utilization of the insured members.

**Funding:** The authors received no specific funding for this work.

**Competing interests:** The authors have declared that no competing interests exist.

## 1. Introduction

National Health Insurance (NHI) programs coverage expansion and membership retention among informal sector workers (ISWs) are significant challenges in most low and lower-middle-income countries (LMICs) [1]. The International Labor Organization (ILO) defines informal employment as all *remunerative* work (i.e., self-employment and wage employment) that is not registered, regulated, or protected by existing legal or regulatory frameworks and *non-remunerative* work undertaken in an income-producing enterprise [2]. These workers do not have secure employment contracts, workers' benefits, social protection, or representation. ILO also reported that in Asia and the Pacific, the non-agricultural, informal employment population is about 60 percent of the total workforce. The percentage increases to about 65 percent if China is excluded and is over 90 percent in some countries. ISWs are generally reluctant to enroll in contributory insurance schemes like social health insurance (SHI) and community insurance, and initiatives to enroll them in such schemes have resulted in adverse selection [3]. Besides, the ILO defines the informal sector as being employers who are unregistered with the government thus unregulated, lacking robust and updated data making it difficult to identify, enroll, and retain NHI members. Many governments also leave them out, focusing first on health insurance for formal-sector workers, followed by government-subsidized enrollment of the poor [4]. Thus, NHI schemes have left out most informal sectors in LMICs, potentially hindering universal health coverage [5].

Numerous studies have identified barriers to informal sector workers' enrolment in NHI. In Ghana, the barriers were the negative influence of traditional risk-sharing arrangements, corruption, shortage of drugs, and politics [6]. While in Indonesia, the high number of household members, financial hardship, membership in other social protection arrangements, lack of comprehensive information about the insurance scheme, and history of previous inpatient care hindered the regular payment of the premium. The Indonesia Social Security Agency for Health (SSAH) reported that nearly 5.5 million (approximately 37%) registered informal sector workers did not regularly pay the premium or became inactive members in 2015 [1]. In Nigeria, age, education, geo-political zone, socio-economic status (SES), and employment status were significant predictors of enrolment in the NHI among women of reproductive age [7]. Apart from the barriers mentioned above, the informal sector, primarily self-employed, faces equity problems while paying the premium [8]. Unlike workers in the formal sector, informal sector workers do not benefit from a contribution split between employer and employee, and the costs of membership are much higher than for formally employed members.

Nepal introduced its NHI policy in 2014 to counteract high Out of Pocket (OOP) health expenditures (55.4% of the health expenditures in 2018) [9] and inadequate government health budget [10]. A semi-autonomous committee was established in 2015 to manage the program and started implementation through piloting in three districts in May 2016. The autonomous Health Insurance Board (HIB) replaced that committee after enacting the NHI Act in 2017. Unlike other countries, Nepal initiated the NHI program with the informal sector, and the Act later integrated the formal sector. The key features of the program include 1. a central pooling mechanism funded through government tax and premium collected through mandatory enrolment (explained below), 2. benefit package of NRs 100,000 (USD$: 841.9; PPP$ 2951.6) [11, 12] for a typical five-member family upon annual payment of NRs 3500 (USD$: 29.5; PPP $: 103.30) premium, and 3. purchase of the health services through public providers using a fee for service (inpatient and diagnostic services) and case-based payment (outpatient and emergency services). The program has been rolled out in 76 districts, out of 77 (based on inquiry to the Health Insurance Board dated September 2021).

The Nepal NHI program faces implementation challenges due to insufficiently defined NHI implementations guidelines, conflicting Act clauses, lack of HIB organizational

guidelines, and inadequate human resources [13]. These challenges have resulted in negative program outputs. A study reported high dropouts of 67%, 44%, and 38% in three consecutive years between 2016 and 2018 [14]. The annual dropout decrement rates were more due to increasing government subsidy for enrollment than increased numbers of members' deciding to remain in the NHI. Other problems included widespread adverse selection, NHI information-related constraints for the members [15], and the low enrolment of poor-households [16]. The NHI Act has authorized mandatory enrolment to reduce dropouts and increase enrolments. However, without comprehensive enforcement guidelines, the enrolment assistants and officers responsible for enrolling members have a daunting task of working out how to find and compel 62.2% of informal workers to enroll [17].

We conducted a semi-structured search using PubMed and google-scholar for reference from studies on NHI members' retention in LMICs and Nepal published from 2015 onwards. The search terms we used included: "NHI", "SHI", "membership retention", "informal sector", "drop-out", "LMICs", and "Nepal". Around 35 articles were retrieved for LMICs, out of which only seven were relevant to the study. Out of seven articles, only two studies investigated sustained premium payments, and the remaining five studies focused on enrolment and expansion. Of the five articles retrieved on Nepal NHI, none were on member retention. The search indicated a one-research gap. Very few studies have emphasized the sustained premium payments in LMICs and there are no studies on members' retention in Nepal.

The proposed study thus aimed to achieve the following objectives in relation to three districts in Nepal:

- Determine the predictors of intention to renew annual subscription to the NHI program among enrolled members.

- Provide information to develop a strategic intervention program through policy-level recommendations, to improve renewals, and strengthen the NHI program

The study will benefit the Nepal NHI program in formulating 'mandatory enrolment' guidelines and prompting large-scale studies of a similar nature. This study will also potentially benefit other LMICs facing high dropouts from NHI schemes in the informal sector.

## 2. Methods

### 2.1 Design

The study employed a concurrent mixed-method study design. The qualitative results have been published elsewhere [13]. This article is based on a quantitative cross-sectional survey.

### 2.2 Setting

The study was based in three districts, namely Bhaktapur, Chitwan, and Kaski, and was purposively selected, meeting inclusion criteria of at least 5% NHI membership enrollment. Additionally, the Principal Researcher; (PR) had easy access to these districts.

### 2.3 Sample size

We used the Cochran 1963 formula to calculate the survey sample size population.

$$n = \frac{Z^2 p(1-p)}{d^2}$$

Where
p = percentage of households facing catastrophic expenditure in Nepal, which is 13.8%

q = 1-p

d = desired level of precision or error allowance

Z = area under the normal curve (from statistical Z table) associated with 95% confidence interval, which is 1.96.

The formula provided a total sample size of 168 participants. Adding a 10% expected dropout rate, the total sample size was increased to 186 participants, allowing for 62 participants in each district. In addition, 61 NHI program dropout respondents were contacted to enquire after their dropout reasons.

### 2.3.1 Inclusion criteria for survey participants.

a. Households (HHs) who have been NHI members and have started their benefits package for at least nine months.

b. HHs who have utilized the service at least once at the time of the survey.

c. Household (HH) heads who dropped out from the NHI program.

d. HHs willing to consent to the study.

**2.3.2 Sampling technique.** We employed a multi-stage sampling technique. Firstly, we selected the districts purposively with 5% NHI membership enrollment and easy access from the capital city. Secondly, we selected the health service centers from the identified districts with at least 1000 NHI members who were willing to support the study. Finally, the eligible HH heads or members attending the health service centers were selected using the simple random method of all insured users registered with the health centers. They were interviewed at the service centers until the sample size was fulfilled.

The district HIB office provided the list of dropout members. The data collectors conveniently selected those residing nearer to them and willing to be interviewed.

## 2.4 Data collection

We employed a close-ended questionnaire to collect the data. The PR developed the new questionnaire after a literature review to identify relevant study variables based on the study objectives. For content validity, the questionnaire was reviewed by three experts on public health and research. The Item-Objective Congruence (IOC) Index was used to evaluate the questionnaire items with + 1 = clearly measuring, 0 = unclear, and -1 = not measuring clearly. A professional translator translated the original English questionnaire into Nepali, then checked back, and finally agreed the questions with the PR. The questionnaire was piloted amongst 45 NHI members who bore similarities to the study participants residing in the study area but was living far from the health centers to avoid cross-contamination. Following pilot testing, we simplified the language, deleted redundant questions, fixed the skip pattern and questions' flow.

The questionnaire included five sections: socio-demography characteristics, becoming an NHI member, the performance of enrollment assistants and officers, benefits package, and replacement of lost card and membership renewal.

We trained university degree holders' data collectors with previous data collection experience. The one-day theoretical and practical training included comprehending the questionnaire (skip pattern, notes), rapport building, and data collection technique. The data collectors conducted face-to-face interviews in the hospital setting after obtaining written informed consent from all participants before the interview.

### 2.5 Variables measurement

**2.5.1 Dependent variable.** The dependent variable was the NHI members' intention to renew their membership (a binary variable coded as 1: yes and 2: no).

**2.5.2 Independent variables.** Andersen's Behavioral Model on Determinants of Health Service Utilization [18] and literature reviews on the factors to enroll and retain NHI membership [6–8] guided the identification of independent variables. The variables were predisposing factors (age, sex, education, family with elderly), enabling factors (occupation, income, NHI information, usage of the benefits package, availability of drugs, satisfaction with the health services, and number of times renewed), and illness level (increased visits after NHI membership, and number of visits to the FPC in the last nine months). Table 1 summarized the independent variables. The variables were recoded into two or three categories.

## 3. Data analysis

The data were analyzed using SPSS version 22. Participants' characteristics were summarized using descriptive statistics. The univariate logistic regression was employed to identify factors crudely associated with the intentions to renew NHI membership. Multivariate logistic regression analysis was used to examine the net effect of the independent variables on the odds of intention to renew. The confidence level was set at 95% and the significance level at 5% (i.e., p-value $\leq 0.05$).

This multivariate regression model included all the independent variables from the univariate analysis that were significant at p-value $\leq 0.20$. Finally, we included the variables that had been found significant in the literature such as age, biological sex, presence of elderly in the HH and occupation [16, 19].

Multi-collinearity testing was conducted using the Variance Inflation Factor (VIF). Age and the presence of elderly persons in the HH were highly correlated with VIFs of value of 24.6 (elderly in the HH) and 25.8 (age). The VIF was less than two after the removal of age from the regression. The Hosmer and Lemeshow chi-square test was employed to test the goodness of fit. and the P-value for this test was $P > .05$, implying good fit.

The questionnaire had one open-ended question on recommendations to improve the NHI program and services. The responses were categorized into seven sub-themes and were used to shed light on the Discussion section.

## 4. Ethics

The PR obtained ethical approval from the Nepal Health Research Council on March 6, 2019 (Reg. # 816/2018) and a no-objection letter from the Health Insurance Board.

The individual consent form outlining study objectives, purpose, respondents' right to withdraw from the interview anytime, and respondent's confidentiality was explained to the participants and written consent was received before the interview.

Only the authors had access to the data.

## 5. Results

### 5.1 Characteristics of the study populations

About 86% of the respondents intended to renew, while 14% did not intend to renew (not tabled). Table 2 provides descriptive statistics of all independent variables. The respondent's mean age was 48 years (SD = 15 years). 56% of the respondents were female, and 46% of the HHs had at least one elderly person. 66.5% of the respondents had attended formal education, while 24% had a college or university-level education. The HH head's employment in the

**Table 1. Summary of study variables.**

| S.N | Variables | Definition of Variables | Variable Description/Coding |
|---|---|---|---|
| | Intention to renew the NHI members | Intentions of the NHI members to renew their annual membership | 1 = Yes |
| | | | 0 = No |
| **Predisposing factors** | | | |
| 1 | Age | Age of the respondents at the time of survey | 1 = 18–49 |
| | | | 2 = 50 and above |
| 2 | Sex | Biological sex of the respondents at the time of survey | 0 = Male |
| | | | 1 = Female |
| 3 | Education | The highest level of education attained by the respondents | 1 = Never went to school but literate |
| | | | 2 = Primary- Higher Secondary |
| | | | 3 = College or university |
| 4 | Family | Family with at least one elderly member | 0 = No |
| | | | 1 = Yes |
| **Enabling factors** | | | |
| 5 | Occupation | Occupation of the head of the HH or any other members | 1 = Formal employment |
| | | | 2 = Informal employment |
| 6 | Income | Monthly income of the respondent's HH | 1 = 10,000–300, 00 NRs |
| | | | 2 = 300,01–100,000 NRs |
| 7 | NHI accurate and clear information | Clarity and accuracy of the information in the first time | 0 = No |
| | | | 1 = Yes |
| 8 | Usage of benefit packages | Usage of benefits package by the HH members in the first year | 1 = Upto 50% |
| | | | 2 = 5–100% |
| 9 | Availability of the drugs | Availability of the drugs on every visit | 0 = No |
| | | | 1 = Yes |
| 10 | Overall satisfaction with the health services | Respondent's satisfaction with health services | 0 = No |
| | | | 1 = Yes |
| 11 | Number of times renewed | Number of times annually renewed by the respondents | 1 = 0 times |
| | | | 2 = 1 time |
| | | | 3 = 2 times |
| **Illness level** | | | |
| 12 | Number of visits to the FPC | Number of times the HH members visited the FPC in the last 9 months | 1 = 1–6 times |
| | | | 2 = 7 and above |
| 13 | Frequency of visits to the FPC | Increased frequency of visits to the FPC after becoming NHI members | 0 = No |
| | | | 1 = Yes |

informal sector was very high (89%). The respondents' overall satisfaction with the health service was very high at 86%; however, only 30.2% increased their facility visits after joining the NHI membership. Similarly, only 18.7% of the respondents used 51–100% of the benefits package.

## 5.2 Variables influencing the intention to renew NHI membership

The univariate analysis (Table 3-unadjusted) showed that HHs with a higher monthly income had lower odds of renewing the NHI membership (unadjusted OR: 0.36; 95% CI: 0.14–0.93). While HHs with overall satisfaction with health (unadjusted OR: 6.2; 95% CI: 2.52–15.23), increased frequency of visits after NHI membership (unadjusted OR: 5.86; 95% CI: 1.33–25.80), and a number of FPC visits in the previous nine months (unadjusted OR: 5.86; 95% CI: 1.33–25.80) had higher odds of renewing their membership.

**Table 2. Characteristics of the study participants (N = 182).**

| Independent Variables | | Frequency | Percentage |
|---|---|---|---|
| Age of the participants | 18–49 years | 98 | 53.8 |
| | 50–100 years | 84 | 46.2 |
| Biological sex | Male | 80 | 44 |
| | Female | 102 | 56 |
| Presence of elderly in the HH | No | 98 | 53.8 |
| | Yes | 84 | 46.2 |
| Education of the participants | Never went to school but literate | 61 | 33.5 |
| | Primary to Higher Secondary | 76 | 41.8 |
| | University | 45 | 24.7 |
| Occupation of HH Head | Formal employment | 20 | 11 |
| | Informal employment | 162 | 89 |
| Monthly income of the HH | 10000 to 30000 NRs | 148 | 81.3 |
| | 30001 to 10000 NRs | 30 | 16.5 |
| NHI information clarity | No | 12 | 6.6 |
| | Yes | 170 | 93.4 |
| Usage of benefit package | Upto 50% | 148 | 81.3 |
| | 51–100% | 34 | 18.7 |
| Availability of drugs in FPC | No | 144 | 79.1 |
| | Yes | 38 | 20.9 |
| Overall health service satisfaction | No | 51 | 13.7 |
| | Yes | 157 | 86.3 |
| No. of times membership renewed | 0 times | 41 | 22.5 |
| | 1 time | 119 | 65.4 |
| | 2 times | 22 | 12.1 |
| No. of FPC visits in last nine months | 1 to 6 times | 127 | 69.8 |
| | More than 6 times | 55 | 30.2 |
| Increased frequency of visits after NHI membership | No | 127 | 69.8 |
| | Yes | 55 | 30.2 |

Out of four variables significantly associated with intention to renew in the univariate analysis, three factors were found significant in the multivariate analysis (Table 3-adjusted result). The HHs with higher monthly income had lower odds of renewing their annual NHI membership (adjusted OR: 0.14, 95% CI: 0.03–0.58)(p = 0.007). Similarly, HHs with overall health service satisfaction (adjusted OR:3.59, 95%CI: 1.23–10.43) and increased frequency of visits after NHI membership (adjusted OR: 10.09, 95% CI: 1.39–73.28) had high odds of renewing their membership.

The variables with high odds to influence the renewal intention, though not statistically significant, were HH head's occupation and HH members who had visited the health facility more than six times. The HH head from the informal sector employment category had lower chances of renewing their membership (adjusted OR: 0.13; 95% CI: 0.01–1.08).

## 5.3 Dropout reasons

Table 4 presents descriptive statistics of the dropout respondents. The mean age of the respondents was 43 (SD = 13.5 years). 62.30% had completed formal education up to Higher Secondary level, and 14.8% had attended university. The informal sector employees constituted 80.3% of the total respondents.

**Table 3. Factors associated with the intention to renew national health insurance.**

| Independent Variables | | Unadjusted results | | Adjusted results | |
|---|---|---|---|---|---|
| | | OR[a] (95% CI) | P-value | OR[b] (95% CI) | P-value |
| Age of the participants | 18–49 years | 1 | 0.507 | 1 | 0.81 |
| | 50–100 years | 1.34 (0.57–3.16) | | 1.02 | |
| Biological sex | Male | 1 | 0.661 | 1 | 0.181 |
| | Female | 1.21 (0.52–2.81) | | 2.28 (0.68–7.62) | |
| Presence of elderly in the HH | No | 1 | 0.507 | 1 | 0.920 |
| | Yes | 1.34 (0.57–3.16) | | 1.07 (0.27–4.24) | |
| Education of the participants | Never been to the school but literate | 1 | | 1 | |
| | Primary to higher secondary | 0.53 (0.17–1.61) | 0.261 | 0.87 (0.17–4.51) | 0.866 |
| | University | 0.36 (0.11–1.15) | 0.085 | 0.51 (0.09–2.86) | 0.447 |
| Occupation of HH Head | Formal employment | 1 | 0.609 | 1 | 0.060 |
| | Informal employment | 0.67 (0.15–3.09) | | 0.13 (0.01–1.08) | |
| Monthly income of HH members | 10000 to 30000 NRs | 1 | 0.034 | 1 | 0.007 |
| | 30001 to 10000 NRs | 0.36 (0.14–0.93) | | 0.14 (0.03–0.58) | |
| NHI information clarity | No | 1 | 0.252 | 1 | 0.479 |
| | Yes | 2.24 (0.56–8.92) | | 1.87 (0.33–10.73) | |
| Usage of a benefit package | Upto 50% | 1 | 0.157 | 1 | 0.387 |
| | 51–100% | 2.94 (0.66–13.14) | | 2.22 (0.36–13.56) | |
| Availability of drugs in FPC | No | 1 | 0.251 | 1 | 0.251 |
| | Yes | 3.42 (0.77–15.21) | | 2.79 (0.48–16.07) | |
| Overall health service satisfaction | No | 1 | <0.001 | 1 | 0.019 |
| | Yes | 6.20 (2.52–15.23) | | 3.59 (1.23–10.43) | |
| No. of times membership renewed | 0 times | 1 | | 1 | |
| | 1 time | 0.47 (0.13–1.71) | 0.254 | 0.54 (0.13–2.23) | 0.393 |
| | 2 times | 0.27 (0.06–1.25) | 0.094 | 0.31 (0.05–2.03) | 0.223 |
| No. of FPC visits in last nine months | 1 to 6 times | 1 | 0.019 | 1 | 0.087 |
| | More than 6 times | 5.86 (1.33–25.80) | | 4.33 (0.80–24.42) | |
| Increased frequency of visits after NHI membership | No | 1 | 0.019 | 1 | 0.022 |
| | Yes | 5.86 (1.33–25.80) | | 10.09(1.39–73.28) | |

a: Univariate logistic regression
b: Multivariate logistic regressions

The top three reasons for dropouts were health services underutilization (43.3%), followed by poor health services (26.9%), and inadequate benefit package (14.9%). The majority of the respondents (95.1%) dropped out after the first year of membership. Almost 64% of the respondents were willing to renew their membership if services were improved.

## 6. Discussion

The study found that some of the enabling factors (HH income and health service quality) and illness level (health service utilization) are the key predictors for Nepal's NHI annual membership renewal. The pre-disposition factors (sex, education, presence of elderly in the HH) though not statistically significant, had some effect on the intention to renew.

Our study results on the enabling factors (HH income and health service quality) are inconsistent with the result from Nigeria [7], Indonesia [1], and Ethiopia [20] where high-income HH members drop out more than poor HH members, and members largely drop out due to poor health service quality. A study result from Vietnam [21] also showed that high-income

**Table 4. Characteristics of the dropout respondents (N = 61).**

| Variables | | Frequency | Percentage |
|---|---|---|---|
| Biological sex | Male | 29 | 47.5 |
| | Female | 32 | 52.5 |
| Presence of elderly in the HH | No | 35 | 57.4 |
| | Yes | 26 | 42.6 |
| Education of the participants | Never been to school but literate | 14 | 22.95 |
| | Primary to Higher Secondary | 38 | 62.30 |
| | University | 9 | 14.8 |
| Occupation of HH Head | Formal employment | 12 | 19.7 |
| | Informal employment | 49 | 80.3 |
| Monthly income of the HH | 10000 to 30000 NRs | 19 | 31.15 |
| | 30001 to 10000 NRs | 38 | 62.30 |
| | I don't know or refused to answer | 4 | 6.56 |
| Membership duration | 1 year | 58 | 95.1 |
| | 2 years | 3 | 4.9 |
| Reasons for dropout | No financial benefit | 4 | 6 |
| | Inadequate benefit package | 10 | 14.9 |
| | Poor health service | 18 | 26.9 |
| | Never utilized it | 29 | 43.3 |
| | Non-affordable premium | 6 | 9 |
| Willing to renew upon improved services | Yes | 39 | 63.9 |
| | No | 3 | 4.9 |
| | I don't know | 19 | 31.1 |

HHs had a high dropout rate. The dropout survey descriptive data depicted that 62.3% of total dropped-out respondents fell into the category of high-income HHs (see Table 4). High-income HHs tend to avoid public health services due to their poor-quality services. Although overall health service satisfaction was high (86.3%), the various aspects of health service delivery were low. For instance, only 18.7% of the respondents used more than 50% of the benefits package, and the drug availability at every visit was 20.9% and the dropout survey supported this finding. Firstly 26.9% of the respondents did not renew due to poor service (the second-highest dropout reason) and lack of drugs. Secondly, almost 64% of dropout respondents were willing to renew their membership upon improved health services.

The other enabling factors (occupation, NHI information clarity, usage of the benefits package, availability of drugs in FPC, and the number of times membership was renewed) though not statistically significant, had influenced the intention to renew. NHI information clarity is part of the NHI design, and respondents with clear information on the enrolment process and benefits package were 1.87 times more likely to renew. The study result from Nigeria [7] depicted that effective NHI design is significantly associated with the enrolment and retention of members. The respondents receiving drugs on every visit were 2.79 times likely to renew, and those who had used the benefits package more than 50% were 2.22 times likely to renew their membership. However, the respondents who had remained long in the program were less likely to renew (OR:0.31 for those renewed twice; OR: 0.54 for those who renewed once). The finding contradicts the decreasing national dropout data from 2016–2018 [14]. However, the same study stated that decreasing dropout was largely due to increasing government subsidy for poor HH enrolment. Since the dropout is high amongst high-income HHs, the study finding remains valid, and is probably due to a lack of improvement in the health service delivery. The qualitative data [13] also stated that large volume of complaints about the NHI

program that were filed to the Health Insurance Board was on account of poor health service delivery.

Our study result stating significant association of dependent variables with increased illness levels was consistent with the results from Nepal [16]. Increased utilization of the health services after NHI membership was used as the proxy variables of illness level in the study. The association was statistically significant (OR: 10.09; p: 0.022). Similarly, respondents who visited the FPC more than six times in the last nine months were four times more likely to renew their membership than those who visited less than six times. The dropout survey further indicated that 40.3%of the respondents did not renew their membership because they never utilized the services (the highest dropout reason). It is inevitable that people with illness tend to visit health facilities more frequently; hence, they intend to renew. However, the other study in Nepal [15] described such health service utilization patterns as adverse selection. The study result has a limitation, as it does not differentiate between moral hazards or actual service utilization, demanding further studies on the health service utilization of the insured members.

Regarding the context of informal sector NHI membership, several studies have mentioned that it is a daunting task to enroll NHI members from the informal employment sector. This study also depicted that, respondents in the informal sector occupation were less likely (OR: 0.13; p: 0.06) to renew, and different studies have substantiated this result [22, 23]. Besides, 80.3% of the dropout respondents were from the informal sector. The NHI Act 2017 has a provision of mandatory enrolment of all Nepalese citizens [24]. However, mandatory enrolment in Nepal could be a challenging task because informal employment constitutes 62.2%, 49% of profit-making companies are unregistered, the informal sector data are often inaccurate and not routinely collected, and because the workers' education levels are low [3]. The enrolment of 11.4% of the unemployed [17] workforce is even more daunting. Historically, the contributory mechanism in the informal sector has been low in many LMICs [25]. All these facts and reality demand a strategic mandatory enrolment guideline and comprehensive implementation mechanism. Until the government meets the mandatory enrolment targets, there is a possibility to cross-subsidize the premium from 50.1% of the registered profit-making enterprises and the formal sector such as government employees.

This study has strengths and limitations. The Nepal NHI program was initiated as a tool to reduce out-of-pocket health expenditure (OOP) [26]. The OOP was as high as 70% of the current health expenditure in the 2000s, which in 2006 led the government to introduce the free health care program with a focus on essential health care services [27]. The NHI has the additional advantage and leverage of proactively reaching out to the poorest, risk-sharing beyond essential health service, and improving the service provision, in addition to health resource mobilization. The study assessing the members' intention to renew NHI membership is a first in Nepal. Despite few studies related to enrollment and dropouts, the knowledge gap is significant in the context of rapid program expansion and evolution. Thus, the study provides insights into the predictors to retain the members and reduce the dropout, particularly amongst the informal sector and poorest group, by improving health services and strengthening the communication/outreach strategy.

The study triangulates the data from two complementary cross-sectional surveys (intention to renew and dropout), reinforcing the result. The study's weaknesses are the small sample size and reduced number of districts participating, making it hard to generalize the result. Despite the small sample size, the results are consistent with studies with bigger sample sizes across LMICs. The study also has not documented the service providers and NHI management perspective, which has been investigated in a qualitative study to be presented elsewhere. Similarly, the 61 dropout respondents were calculated without statistical formula. However, the benefit of collecting information from these respondents outweighs the limitation and has

enriched the results and discussion by triangulating the membership renewal predictors from two different groups.

The study suggests two types of recommendations. Firstly, the overall suggestions, based on the study result, increase retention and reduce dropout, including future studies, and are as follows.

1. Given the difficulty with enrolling informally employed populations and high dropout, HIB should develop a strategic guideline to implement the NHI mandatory enrolment with a specific focus on the informal sector populations.

2. As found, NHI information clarity influences the member's intention to renew, HIB should regularly assess the NHI communication outreach and its impact, emphasizing the informal sector populations and also develop a different communication strategy focusing on the informal sector populations.

3. Given sparse NHI-related studies in Nepal, HIB should organize national conferences and seminars with diverse stakeholders for consistent demand, knowledge generation, and affirmative policy changes.

Secondly, respondents provided recommendations to improve health service delivery. A total of 250 recommendations were collected, which were divided into seven themes as follows.

4. Improve drug availability at FPC: The prescribed drugs should be available in FPC at every visit. In particular, the expensive drug should be included in the benefits package. If such expensive drugs are not included, the patients should be informed before prescribing such drugs.

5. Have all listed services available: The FPC should increase the facilities and services to include as many services as listed in the benefits package.

6. Improve attitude of health workers: Health workers should be polite and not discriminate against the insured populations. They should be regularly trained in supporting NHI members.

7. Improve hospital management for enabling environment to access the services: The enabling environment could include crowd management and reduce waiting time, and people's chart with detailed NHI information for insured population, increased registration time duration for the insured population, improvement of emergency and operations facilities, appropriate and friendly services for elderly and disabled, improve the referral system by referring to hospital with better facilities.

8. Improve technical software and benefits package information: The software should enable a swifter registration process at FPC. The benefits package usage information should be easily accessible.

9. Increase the number of health workers at FPC.

10. The need for more hospitals: Increase the number of hospitals, including the private hospitals, to offer more options for the members to choose the FPC. Additionally, more service centers would reduce the travel time and motivate members.

## 7. Conclusion

The study found some key Nepal NHI annual membership renewal predictors, including HH income, health service quality, and health service utilization. Given that different LMICs,

including Nepal, have faced challenges in enrolling and retaining members from the informal employment sector, the study offers insights to strengthen the NHI program in increasing their members. The NHI program has now expanded to all districts in Nepal. These research findings could form the basis of developing systems and processes to increase NHI program membership in Nepal. An improved system with adequate retention as well as the development of training and support structures for healthcare professionals will prevent the decline of financial resourcing for the program, otherwise will threaten NHI sustainability in the future.

## Supporting information

**S1 File. Inclusivity in global research.**
(DOCX)

**S2 File. Questionnaire for the respondents who did not renew their members.**
(DOCX)

**S3 File. Survey questionnaire with the insured populations: Process evaluation.**
(DOCX)

**S1 Data.**
(XLSX)

**S2 Data.**
(XLSX)

## Acknowledgments

The author acknowledges the institutional support of the College of Public Health Sciences, Chulalongkorn University to complete this study. The author is grateful to the Health Insurance Board for the non-objection letter to conduct the study. We are also equally grateful to all the study participants who provided ample time for interview, and to the data collectors for their contributions in data collection.

Finally, the author is highly thankful to Dr. Calistus Wilunda of the African Population and Health Research Center for his useful comments and suggestions on the manuscript.

## Author Contributions

**Conceptualization:** Gaj Bahadur Gurung, Alessio Panza.

**Data curation:** Gaj Bahadur Gurung.

**Formal analysis:** Gaj Bahadur Gurung, Alessio Panza.

**Investigation:** Gaj Bahadur Gurung.

**Methodology:** Gaj Bahadur Gurung, Alessio Panza.

**Project administration:** Gaj Bahadur Gurung.

**Resources:** Gaj Bahadur Gurung.

**Software:** Gaj Bahadur Gurung.

**Supervision:** Gaj Bahadur Gurung.

**Validation:** Gaj Bahadur Gurung.

**Visualization:** Gaj Bahadur Gurung.

**Writing – original draft:** Gaj Bahadur Gurung.

**Writing – review & editing:** Gaj Bahadur Gurung, Alessio Panza.

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
