## [Decision Letter · Decision Letter 0]

15 Dec 2021

PGPH-D-21-01035

Predictors of annual membership renewal to increase the sustainability of the Nepal National Health Insurance program: a cross-sectional survey

Dear Mr. Gurung,

Thank you for submitting your manuscript to PLOS Global Public Health. After careful consideration, we feel that it has merit but does not fully meet PLOS Global Public Health’s publication criteria as it currently stands. Therefore, we invite you to submit a revised version of the manuscript that addresses the points raised during the review process.

We look forward to receiving your revised manuscript.

Kind regards,

Nnodimele Onuigbo Atulomah, PhD

Academic Editor

Journal Requirements:

1. Please include a complete copy of PLOS’ questionnaire on inclusiveness in global research in your revised manuscript. Our policy for research in this area aims to improve transparency in the reporting of research performed outside of researchers’ own country or community. The policy applies to researchers who have traveled to a different country to conduct research, research with Indigenous populations or their lands, and research on cultural artifacts. The questionnaire can also be requested at the journal’s discretion for any other submissions, even if these conditions are not met.  Please find more information on the policy and a link to download a blank copy of the questionnaire here: https://journals.plos.org/globalpublichealth/s/best-practices-in-research-reporting. Please upload a completed version of your questionnaire as Supporting Information when you resubmit your manuscript.

3. Please update the completed 'Competing Interests' statement, including any COIs declared by your co-authors. If you have no competing interests to declare, please state "The authors have declared that no competing interests exist".

4. Please amend your Data Availability Statement and indicate where the data may be found

Additional Editor Comments (if provided):

The three reviewers have considered the manuscript and made their comments for your considerations. This study is very important because of the central subject of consideration relates to access to health care, universal coverage, and burden of out-of-pocked borne by population seeking health care services in the context of public health as a population science. Importantly, factors associated with barriers to NHI enrollment have been identified from literature reviewed.

The abstract needs a statement of objective to indicate the course of action and direction of the study. The statement describing sampling in the study would have explained how a sample of 182 participants were drawn from a larger mixed-method study. There is observed inadequate presentation of the methodology and results in the abstract which needs modification.

There is a need to modify the study objective to read; “This study sought to determine predictors of intention to renew annual subscription for NHI program among enrolled members, provide information to develop strategic intervention programme, through policy level recommendations, to improve renewals and strengthen the NHI programme. The rest of the statements in the last sentence should be deleted.

The statistical analysis and presentation of results are flawed. For instance, the tables appear busy/crowded with data.

Kindly review the comments of the reviewers carefully and make necessary modifications that would make the manuscript acceptable.

Reviewers' comments:

Reviewer's Responses to Questions

**Comments to the Author**

1. Does this manuscript meet PLOS Global Public Health’s publication criteria? Is the manuscript technically sound, and do the data support the conclusions? The manuscript must describe methodologically and ethically rigorous research with conclusions that are appropriately drawn based on the data presented.

Reviewer #1: Yes

Reviewer #2: Yes

Reviewer #3: No

2. Has the statistical analysis been performed appropriately and rigorously?

Reviewer #1: Yes

Reviewer #2: No

Reviewer #3: Yes

3. Have the authors made all data underlying the findings in their manuscript fully available (please refer to the Data Availability Statement at the start of the manuscript PDF file)?

Reviewer #1: Yes

Reviewer #2: Yes

Reviewer #3: Yes

4. Is the manuscript presented in an intelligible fashion and written in standard English?

Reviewer #1: Yes

Reviewer #2: Yes

Reviewer #3: No

5. Review Comments to the Author

Reviewer #1: The manuscript presents an investigation of Predictors of annual membership renewal to increase the sustainability of the Nepal National Health Insurance program among informal sectors using mixed methods. The manuscript is generally well written and presented.

Reviewer #2: The title sounds good and reflects the context of the paper. The author gave all necessary background information but can also be improved upon. The methodology approach is good. Sample size computation and sampling technique is scientifically grounded. However, the analysis carried out and the conclusion should be checked. The author mentioned chi square to test relationship while simple correlation Pearson will establish relationships between the independent variable and dependent variables. Also the predicting variables were be clearly stated which connects with the title.. For instance, three factors such as income, health quality and health service utilisation were stated as predicting factors for membership renewal of health insurance.

Reviewer #3: The authors have chosen a very interesting and relevant topic for Nepal. However, the structure of paper is not in PLOS GLOBAL Health format and writing style is very poor. The abstract has not mentioned objective of the study as well as conclusion of the study. Authors have mentioned that "The program (NHI) has been rolled out in 75 districts, out of 76 (Health Insurance Board based on inquiry dated September 2021)". However, Nepal has 77 districts. Many sentences are incomplete such as "We conducted a semi-structured search using PubMed and google-scholar for reference from

studies on NHI members’ retention in LMICs and Nepal published in 2015 onwards". They have also mentioned that "The proposed study thus aims to achieve the following objectives in Nepal:

Analyze the factors affecting the NHI members’ intention to make annual renewals in

the program

Provide concrete programmatic and policy level recommendations to strengthen the

program" It seems like proposal of the study.

Importantly, authors have mentioned that they have employed a concurrent mixed-method study design. The qualitative results have been published elsewhere. In my opinion, publishing findings of mixed method study separately is practice of salami publication which is not acceptable. The discussion section of the paper is also poorly done. Hence, I recommend rejection of this paper.

6. PLOS authors have the option to publish the peer review history of their article (what does this mean?). If published, this will include your full peer review and any attached files.

**Do you want your identity to be public for this peer review?** For information about this choice, including consent withdrawal, please see our Privacy Policy.

Reviewer #1: **Yes: **Titilayo Olaoye

Reviewer #2: **Yes: **Dr Adebayo

Reviewer #3: No

---

## [Editor Report · Decision Letter 1]

13 Feb 2022

Predictors of annual membership renewal to increase the sustainability of the Nepal National Health Insurance program: a cross-sectional survey

PGPH-D-21-01035R1

Dear Dr. Gurung,

We are pleased to inform you that your manuscript 'Predictors of annual membership renewal to increase the sustainability of the Nepal National Health Insurance program: a cross-sectional survey' has been provisionally accepted for publication in PLOS Global Public Health.

Best regards,

Nnodimele Onuigbo Atulomah, PhD

Academic Editor

After considering the modifications made following reviewers recommendations, I am reporting that these modifications meets the expectations of the editors and recommends that the manuscript be advanced to the next stage in the editorial process.